# Characterization of De Novo Promoter Variants in Autism Spectrum Disorder with Massively Parallel Reporter Assays

**DOI:** 10.3390/ijms24043509

**Published:** 2023-02-09

**Authors:** Justin Koesterich, Joon-Yong An, Fumitaka Inoue, Ajuni Sohota, Nadav Ahituv, Stephan J. Sanders, Anat Kreimer

**Affiliations:** 1Center for Advanced Biotechnology and Medicine, Rutgers University, Piscataway, NJ 08854, USA; 2Department of Cell and Developmental Biology, Rutgers University, Piscataway, NJ 08854, USA; 3Department of Psychiatry and Behavioral Sciences, Weill Institute for Neuroscience, University of California, San Francisco, CA 94143, USA; 4School of Biosystem and Biomedical Science, College of Health Science, Korea University, 145 Anam-ro, Seongbuk-gu, Seoul 02841, Republic of Korea; 5BK21FOUR R&E Center for Learning Health Systems, Korea University, Seoul 02841, Republic of Korea; 6Department of Bioengineering and Therapeutic Sciences, University of California, San Francisco, CA 94158, USA; 7Institute for Human Genetics, University of California, San Francisco, CA 94158, USA; 8Institute for the Advanced Study of Human Biology (WPI-ASHBi), Kyoto University, Kyoto 606-8501, Japan; 9Institute for Developmental and Regenerative Medicine, Old Road Campus, Roosevelt Dr, Headington, Oxford OX3 7TY, UK; 10Department of Biochemistry and Molecular Biology, Robert Wood Johnson Medical School, Rutgers University, Piscataway, NJ 08854, USA

**Keywords:** functional genomics, gene regulation, computational genomics, neural development, autism disease genetics

## Abstract

Autism spectrum disorder (ASD) is a common, complex, and highly heritable condition with contributions from both common and rare genetic variations. While disruptive, rare variants in protein-coding regions clearly contribute to symptoms, the role of rare non-coding remains unclear. Variants in these regions, including promoters, can alter downstream RNA and protein quantity; however, the functional impacts of specific variants observed in ASD cohorts remain largely uncharacterized. Here, we analyzed 3600 de novo mutations in promoter regions previously identified by whole-genome sequencing of autistic probands and neurotypical siblings to test the hypothesis that mutations in cases have a greater functional impact than those in controls. We leveraged massively parallel reporter assays (MPRAs) to detect transcriptional consequences of these variants in neural progenitor cells and identified 165 functionally high confidence de novo variants (HcDNVs). While these HcDNVs are enriched for markers of active transcription, disruption to transcription factor binding sites, and open chromatin, we did not identify differences in functional impact based on ASD diagnostic status.

## 1. Introduction

Autism spectrum disorder (ASD) is a highly heritable, complex neurodevelopmental disorder with a genomic architecture that includes both common [1] and rare/de novo [2] variants (DNVs) in the coding [3] and non-coding genome [4]. While the DNV contribution in coding regions has been well characterized in ASD, there has been limited insight into the extent to which non-coding DNVs impact neurodevelopment [5,6], especially in promoter and enhancer regulatory regions [1]. An analysis of whole-genome sequencing data in a cohort of almost 2000 quartet families (composed of both parents, a child with ASD, and an unaffected sibling control) implicated DNVs in promoter regions 1–2000 bp upstream of the transcription start site (TSS) of genes [7,8]. Exploratory analysis showed that the promoter DNVs observed in ASD cases were more likely to be at nucleotides conserved across species and at transcription factor binding sites (TFBSs) than those in unaffected sibling controls. We hypothesized that non-coding promoter variants contribute to neurodevelopmental phenotypes by disrupting the transcriptional regulation of the nearby gene and, subsequently, the encoded protein.

Here, we followed up on the promoter region results found by An et al. [7] by examining the transcriptional impacts of DNVs in neural progenitor cells (NPCs). Since tissues and cell type influence gene expression [9,10,11,12], cell type is likely to impact the functional consequences of DNVs. NPCs have been implicated with ASD and similar neurodevelopmental disorders [13,14,15,16], though to a lesser degree than maturing neurons. We assessed epigenetic data generated from NPCs to demonstrate that genes previously associated with neurodevelopmental delay that have a DNV in their promoter are enriched for open chromatin [17], are markers of active transcription [18] at their promoters, and are transcribed [19].

NPCs are also tractable for the lentivirus-based massively parallel reporter assay (lentiMPRA) [20], which can simultaneously test thousands of sequences in a single experiment. These sequences included both wildtype (reference) and mutated (alternate) forms upstream of a minimal promoter and reporter gene, so that changes in the expression of the reporter gene can be detected for each sequence.

Using this approach, we characterized 3600 promoter DNVs in ASD cases and sibling controls. A subset of 165 high confidence DNVs (HcDNVs) was identified based on the difference between reference and alternate reporter gene expression (false discovery rate (FDR) ≤ 0.2 and absolute differential expression ≥ 0.1). These HcDNVs were enriched with transcriptionally related genomic annotations including transcription factor binding and epigenetic markers of active transcription, suggesting they play a role in gene regulation. These insights into predictors of functional effect in promoter regions might inform future efforts to identify the association between non-coding rare variants and neurodevelopmental disorders, or to design similar functional analyses of such variants.

If a subset of promoter DNVs mediate risk for ASD, we would expect to observe a higher proportion of DNVs with a functional impact in ASD cases than controls, or a greater degree of functional impact in DNVs in ASD cases than controls. Our primary analysis did not identify such differences. Based on these results, we recommend that future functional analyses include larger numbers of variants; are performed in cell types with stronger evidence for a role in ASD, such as excitatory neurons; and, as performed in this study, ascertain assay variants consistently in both cases and controls.

## 2. Results

### 2.1. ASD-Associated DNVs in Conserved Promoter Regions Overlap with NPC Epigenetic Signatures

Cell type is a critical variable in non-coding functional assays because the regulatory machinery, including chromatin accessibility, transcription factors, and levels of gene expression, differ between cell types [21]. NPCs are human-derived cells that are well-characterized and have been previously used for large-scale functional assays that model aspects of the gene regulatory environment of the developing human brain. To assess the suitability of NPCs as a gene regulatory model of promoter regions in ASD, we analyzed whether 174 ASD-associated genes [22] that have DNVs in their promoter were enriched for accessible chromatin and epigenetic markers of active transcription [23], using NPC-derived assay for transposase accessible chromatin (ATAC-seq) data and acetylation of lysine 27 of histone 3 (H3K27ac) chromatin immunoprecipitation (ChIP-seq) data previously generated in our lab [24]. Protein-coding genes were used as the background, excluding 817 genes associated with neurodevelopmental delay [22] and ASD genes (non-ASD genes). Both the ASD and non-ASD gene lists were filtered to include only genes with at least one of the 6787 promoter DNVs identified in cases or controls by An et al. [7] and genes with a probability loss-of-function intolerance (pLI) score ≥ 0.90 [25], so that both gene lists would be similarly intolerant to mutations. We found that ASD genes with a promoter DNV were more likely to be near both ATAC-seq and H3K27ac ChIP-seq peaks than non-ASD genes with a promoter DNV (*p* = 7.5 × 10^−33^, 7.4 × 10^−41^, Odds Ratio = 4.89, 12.7, respectively, two-sided FET), lending support to the notion that NPCs might be a useful model for observing the functional impact of de novo mutations observed in ASD.

### 2.2. Selection of DNVs for lentiMPRA

We next set out to characterize the effect of promoter DNVs on gene expression in a high-throughput manner using lentiMPRA. We selected 3600 (1888/1712 cases/controls) DNVs that are in promoters, defined as ≤2000 bp upstream of a transcription start site [7], using the following subsets: (1) all 923 (520/403 cases/controls) DNVs at evolutionarily conserved loci based on PhyloP and PhastCons score (Section 4.2) and (2) 2677 randomly selected DNVs (1368/1309 cases/controls) from the 3672 DNVs prioritized by the DAWN hidden Markov random field model [26] or the LASSO regression model [27] that were previously used to predict ASD status from annotation categories [7] but not at evolutionarily conserved loci. For each of the 3600 promoter DNVs, we generated a 200-nucleotide “alternate allele sequence (alternate)” with the variant in the middle surrounded by the GRCh38 reference sequence and the corresponding “reference allele sequence (reference)” with the variant replaced by the GRCh38 reference sequence (Figure 1A). For deletions, we created a wildtype sequence of 200 nucleotides and a shorter variant sequence due to the deleted nucleotides. For insertions, the variant sequence was 200 nucleotides, with a shorter wildtype sequence. We also generated a set of 150 “positive control sequences” based on ATAC-seq, H3K27ac ChIP-seq, and RNAseq signals in NPCs (Figure 1B; Methods) and 150 “negative control sequences” which included 200 nucleotide scrambled sequences of 75 case DNVs and 75 control DNVs selected at random (Figure 1C). Altogether, this resulted in a library of 7500 DNA sequences (3600 reference, 3600 alternate, 150 Positive, and 150 Negative).

### 2.3. LentiMPRA Identifies Functional Variants within NPCs

The 7500 DNA sequences were synthesized and cloned upstream of a minimal promoter (mP) into the lentiMPRA vector (Figure 1D;**Section 4.8**). During the cloning process, 15-bp random barcodes were placed in the 5′UTR of the EGFP reporter gene [28]. The association between the cloned sequences and barcodes was determined via DNA-seq (Section 4.9). Lentivirus was generated and NPCs were infected with the library (Figure 1D). Four days post infection, cells were harvested, and DNA barcodes and transcribed RNA barcodes were quantified by DNA-seq and RNA-seq, respectively (Figure 1D). The library infections were carried out using three biological replicates.

Using MPRAflow, a computational pipeline developed in our group [20] (Figure 1E), we took a stringent approach to associate barcodes with the cloned sequences. For each barcode, we required at least 80% of the reads associated with the barcode to map to a single sequence, with at least three reads supporting that assignment, resulting in over 875,000 confidently assigned barcodes and averaging 125 barcodes per library sequence (Methods). We then analyzed the barcodes from the lentiMPRA-infected cells and matched them with the assigned barcodes of the library. Across biological replicates, we were able to confidently assign an average of 47% of the barcodes (Section 4.9; Appendix A). Considering only confidently assigned barcodes that had a representation both in RNA and DNA from infected cells, we observed an average of 121 barcodes per library sequence in each replicate (Appendix A), corresponding to 3090 out of the 3600 variant pairs (i.e., both variant and wildtype sequences), 144 positive control sequences, and 149 negative control sequences. (Appendix A).

We then used MPRAnalyze [29] (Figure 1E) to aggregate the barcodes and estimate “alpha”, the transcription rate induced by each tested sequence. We observed reproducible results between replicates (average Pearson correlation 0.98; Appendix A). As expected, the median alpha value was highest for the positive control sequences and lowest for the negative control sequences. The median alpha of the wildtype and variant sequences was marginally greater than the negative control sequences (Figure 2A,B). We found a total of 487 active reference alleles and 473 active alternate alleles when empirically quantifying the alpha values compared to the negative controls.

### 2.4. Delineating DNVs’ Effects

We next quantified the magnitude of deviation between the wildtype and variant sequence transcription rates (natural log(wildtype alpha/variant alpha)) and the corresponding confidence interval using false discovery rate (FDR). Comparing the MPRA signal of the 3090 DNVs with results for both the wildtype and variant sequences (Section 4.3), we defined the threshold for high confidence variants (HcDNV) as FDR ≤ 0.2 and absolute differential expression ≥0.1 between the wildtype and variant sequences. The majority of DNVs (2925 DNVs, 94.7%) did not meet this threshold and are referred to as ‘background’ variants. The remaining 165 HcDNVs were split approximately evenly between ‘activator’ loci, in which the DNV reduced expression (wildtype alpha > variant alpha), and ‘inhibitor’ loci, in which the DNV increased expression (wildtype alpha < variant alpha) (Figure 2C; Appendix A). Analyses were additionally conducted with a lower (<0.10) FDR threshold for the HcDNVs but this did not alter any of our conclusions. Similar patterns of enrichment as shown in the analyses below were found but had a larger *p*-values, likely due to the smaller size of the HcDNV grouping (122/165) reducing the statistical power. We choose to showcase the results of the HcDNVs using the FDR < 0.20 and differential expression thresholds to maintain a higher level of statistical power in our analyses.

### 2.5. No Evidence of Association between ASD Diagnosis and Functional Impact

To test our primary hypothesis that a higher proportion of mutations in ASD cases would have a functional impact than those in controls, we assessed whether the proportion of DNVs observed in ASD cases and sibling controls differed between the HcDNVs and the 2925 background DNVs. We did not observe a difference between 78 of the 165 HcDNVs in ASD cases (47.2%) and 1554 of the 2925 background DNVs (53.1%), *p* = 0.14, FET, OR = 0.79, 95% CI = 0.57–1.10. Furthermore, in the HcDNVs, we did not observe a difference in effect size using either log fold change (log(wildtype alpha/variant alpha), *p* = 0.91 Wilcoxon rank sum test), deviation of the variant sequence alpha value from wildtype sequence alpha (abs(wildtype alpha − variant alpha)/variant alpha, *p* = 0.08, Wilcoxon rank sum test), or FDR value (*p* = 0.47, Wilcoxon rank sum test). We did observe a nominally significant (*p* = 0.016 Wilcoxon rank sum test) greater functional impact for HcDNV inhibitors in ASD cases than controls (Figure 2D–F), though this did not survive correction for the four equivalent conditions, so does not constitute evidence of a functional effect based on ASD diagnostic status (Appendix A).

Considering the full list of 3090 DNVs, we did not see evidence of higher functional impact in DNVs from ASD cases in the full list of 3090 DNVs based on log fold change (*p* = 0.32 Wilcoxon rank sum test) or FDR value (*p* = 0.20, Wilcoxon rank sum test) (Appendix A). We also observed similar proportions of ASD cases and controls across subgroups of variants, including only activators, only inhibitors, and only background DNVs (Appendix A; Appendix A). As an exploratory analysis, we assessed whether the genes in which the promoter HcDNVs were enriched for the 72 genes associated with ASD or the 373 associated with neurodevelopmental delay [22]. We did observe such an enrichment (Appendix A).

### 2.6. High-Confidence DNVs Are Enriched in Regulatory Categories and Epigenetic Marks

To help guide future functional characterization analyses of variants in regulatory regions, we sought to characterize the HcDNVs irrespective of their involvement in ASD to understand what factors predict the functional impact of a DNV in NPCs. To this end, we compared the 165 HcDNVs to the 2925 background DNVs. We first utilized the 28 annotation categories in promoter regions reported by An et al. to see which are enriched in the HcDNVs (Methods; Appendix A). Correcting for 28 comparisons, we observed ten significant categories in two functional groups, as described by An et al.: (1) markers of active transcription (‘Active TSS’) and (2) measures of conservation across species (‘Conserved Loci’) (Figure 3A). The Active TSS group includes seven annotation categories: ENCODE transcription factor binding sites (*p* = 1.5 × 10^−16^, FET), Roadmap Epigenome DNase sites (*p* = 6.1 × 10^−15^, FET) [30,31], chromatin state 1–flanking active TSS [30] (*p* = 1.6 × 10^−12^, FET), ENCODE DNase sites (*p* = 1.2 × 10^−11^, FET), mid-fetal H3K27ac peaks (*p* = 1.8 × 10^−7^, FET), DAWN cluster C18 (*p* = 4.7 × 10^−7^, FET), and, finally, the combined ‘Active TSS’ group (*p* = 1.1 × 10^−5^, FET). The Conserved Loci group includes PhyloP [32] (*p* = 6.1 × 10^−4^, FET), DAWN cluster C20 (*p* = 6.1 × 10^−4^, FET), and DAWN cluster C63 (*p* = 0.0011, FET).

Given the enrichment of DNase and H3K27ac, we considered whether the NPC-derived epigenetic markers (Section 4.1) were similarly correlated with functional impact. Using the same approach of assessing HcDNV enrichment vs. background DNVs, both ATAC-seq (*p* = 5.4 × 10^−18^, FET) and H3K27ac ChIP-seq (4.0 × 10^−8^, FET) peaks were enriched for HcDNVs (Figure 3B).

### 2.7. High-Confidence Variants Are Significantly Enriched for Upstream and Downstream Disruption of Transcription Factors

Having established the enrichment of HcDNVs for active transcription and TFBS (Figure 3A), we wanted to identify whether the DNVs were disrupting TFBSs and if they can disrupt other genes’ transcription by being upstream of a gene that encodes a transcription factor. To identify TFBSs that are interacting with and possibly disrupted by the DNV, we used ‘Find Individual Motif Occurrences’ (fimo) ([33]; Section 4.1) to identify transcription factor motifs and restricted the results to motifs that included the DNV site. Comparing the reference sequence of HcDNVs and background DNVs, we observed substantial enrichment for fimo-defined TFBS in the HcDNV sequences (3.32 motifs per HcDNV vs. 1.23 motifs per background DNV, *p* = 4.9 × 10^−30^, FET, Figure 4A). Repeating the motif discovery analysis using the variant alleles instead, we found that about half the motifs are disrupted by DNVs in the background variants, but three-quarters of the motifs are disrupted in the HcDNVs *p* = 1.2 × 10^−16^, FET) (Figure 4B, Appendix A). Among the motifs that were being disrupted, we found core promoter motifs YY1, HEY1, and a master negative regulator for neurogenesis REST.

Our motif analysis suggests that a substantial fraction of the functional impact of DNVs might be through the disruption of transcription factor binding sites. We next considered whether the 165 HcDNV promoter sequences were upstream of transcription factors themselves, such that a small change in expression might have a large impact distributed across numerous regulated genes. Utilizing gene set enrichment analysis (GSEA) [34] (**Methods**), we found that the ‘transcription regulator complex’ and the gene targets ZNF740, KAT2A, BRCA2, SKIL, and REPIN1 were enriched (Figure 5A). Similarly, we assessed whether there was evidence of the HcDNV gene targets converging on protein complexes or networks by using DOMINO [35] (Section 4.1) to assess the enrichment of protein–protein interactions and gene ontology terms. We saw one small subnetwork of nine interacting transcription factor genes, which adds evidence to the possibility of a domino-like disruptive effect on transcription caused by these variants (Figure 5B).

### 2.8. HcDNVs Have Enriched GC Content and Top-Scoring Variants Show Similar Patterns to Overall HcDNV Enrichments Found

Regulatory regions that affect gene expression are often GC rich [36]. To assess this pattern in the HcDNVs, we calculated the percentage GC across the 200-nucleotide library sequence using the reference allele. We found that the average GC content of the background variants is similar to the average GC content of the human genome by approximately 50%, while the GC content of the HcDNVs is higher at ~63% (*p* = 3.9 × 10^−36^, Wilcoxon rank sum test, Appendix A). Finally, we wanted to highlight two HcDNVs to show that irrespective of ASD involvement, the MPRA assay can identify variants that are deeply involved in gene regulation and can cause large disruptions to gene regulation. Both presented variants overlap the ATAC-seq and H3K27ac ChIP-seq NPC peaks, are found in active transcription start sites, are conserved loci, and are found in the C18 DAWN cluster, and the ENCODE TFBS, ENCODE DNase, and REP DNase annotation categories, along with disrupting genes that have a pLI score > 0.90. The first DNV, at human genome build 38 (hg38) coordinates chr11:76445841 C>T, originated in an ASD control sibling, had an FDR value of 2.0 × 10^−4^, and disrupted the gene *EMSY* (Figure 6A). *EMSY* encodes a protein that has been predicted to be involved in DNA repair by way of regulating transcription. The other DNV, at human genome build 38 (hg38) coordinates chr19:8005629 C>A, originated in an ASD case child, had an FDR value of 1.0 × 10^−5^, and disrupted the gene *ELAVL1* (Figure 6B). *ELAVL1* encodes a protein that recognizes mRNAs and degrades them as a means of gene expression regulation.

## 3. Discussion

In this study, our goal was to further characterize de novo variants found in the promoter region of ASD families using lentiMPRA to functionally test the changes in transcription due to these variants. As found by An et al. in 2018 [7], there was a small enrichment for ASD DNVs originating from children with ASD vs. sibling controls, especially at sites of evolutionary conservation. However, this enrichment was only found after developing a DNV risk score and comparing different annotation categories using a LASSO regression model to find significant annotations that lead them to this finding, prompting us to examine this further with functional data. Prior MPRA analyses of non-coding regulatory regions that have well-documented associations with Mendelian disorders in disease-relevant cell lines showed that variants in promoter regions can have a detectable functional impact on gene expression [37].

Here, we used lentiMPRA to assess the functional impact of 3600 promoter *DNVs* selected from ~250,000 total DNVs in 3804 ASD cases and unaffected sibling controls. The functional consequences in NPCs were minimal for the majority of DNVs, with only 5% (165 DNVs) meeting our high-confidence threshold for functional impact, and 15.8% of reference alleles and 15.3% of alternate alleles achieving significantly higher alpha values compared to negative controls. We did not see clear evidence of enrichment for DNVs from ASD cases in this high confidence set or evidence of increased functional impact from ASD-derived DNVs across all variants. There are many possible explanations for these negative results, including the choice of cell type (recent analyses have implicated developing neurons rather than NPCs in ASD [4]), the number of DNVs tested, the depth of sequencing, the relatively small size of the promoter region used to test transcription function, the absence of genomic context, and the limited evidence of ASD association in the initial WGS discovery cohort. To address some of these limitations, we aim to incorporate these variants and build upon the design that was tested here to both address these limitations and compare the results in different cell types and/or different genomic regions. In addition, this study is limited by MPRA experiment to study individual mutations and did not include possible aggregate disruptions. Future studies are being planned to develop computational models based on MRPA findings that may predict the effect of disruption due to aggregate mutations.

While our results do not provide clear insights into ASD, they do provide insights into the potential for DNVs to alter gene expression and the factors that predict such changes. We found that these HcDNVs frequently disrupt the binding of transcription factors, providing a mechanism for the altered levels of transcription. Furthermore, we found that the gene products affected by these variants are themselves involved in transcription factor binding and could cause a cascade effect to disrupt the transcription level of many downstream genes. This gives us an additional foundation for the belief that MPRA is a reliable model for high-throughput functional testing of variants to find which ones are causing significant transcriptional regulation, and that more studies will need to be conducted to find which variants are utilizing this mechanism to cause the disruptions seen in complex human disorders. We also note that newer studies are highlighting excitatory and cortical neurons as a more likely source of ASD disruptions than NPCs [38], and while these insights were less clear when designing this experiment, they will be very beneficial in developing our next steps, where we aim to study these DNVs in other cell types to hopefully see better results.

With the hope of guiding future efforts to functionally characterize variants identified in neurodevelopmental disorders, we recommend that future functional assays include variants ascertained in both cases and controls, perform assays in excitatory and/or inhibitory neurons, assess more variants to increase statistical power, include additional replicates and greater sequencing depth to improve detection of functional effects, and consider genomic context to identify possible disruptions to additional regulatory elements that may confound transcriptional dysregulation.

## 4. Materials and Methods

### 4.1. Computational Analysis

#### 4.1.1. Computational Pipeline for ChIP-seq and ATAC-seq

For both ChIP-seq and ATAC-seq, we used the FASTQC pipeline (Version 0.3.2; Babraham Bioinformatics) on our reads and aligned them to the reference genome (hg19) with bowtie version 1.1.1 [39] for ChIP-seq, and bowtie2 version 2.2.9 [40] for ATAC-seq, retaining only reads that mapped to a unique position in the genome [“–m 1”]. We marked duplicate reads in the bam files using PICARD and checked for the contamination of primer sequences using Trimmomatic (Version 0.3.2) [41]. For each of our H3K27ac ChIP-seq replicate pairs, peaks were called using MACS2 version 2.1.0 [18], with the relevant control input file with default parameters (setting the FDR to 0.05 and default hg19 human genome size). We intersected the peaks from the two replicates using bedtools [42]. For ATAC-seq, after the alignment of the reads to the reference genome, reads aligned to the positive strand were moved by +4 bp, and reads aligning the negative strand were moved by 5 bp. For each of our two replicate pairs, we called peaks using MACS2 version 2.1.0 [18] with default parameters (setting the FDR to 0.05 and default hg19 human genome size). We intersected the narrow peaks from the two replicates using bedtools [42]. To achieve a similar number of peaks found in NPCs and ESCs, we applied the following filters to the NPC epigenetic peak files: a log fold change > 4 between the peak compared to background input filter and merged peaks within 1000 bp for H3K27ac peaks, and a log fold change > 6 between the peak compared to background input with a minimum length of the peak being a 250-bp and merged peaks within 100 bp.

#### 4.1.2. ASD Gene Set Analysis

The ASD gene list was taken from Fu et al., 2021 [22], which contains 817 genes associated with ASD and developmental delay. Grep was then used to find genes from this list that were present as the closest gene to any of the promoter variants from An et al. This then allowed us to generate a list of ASD genes and a list of non-ASD genes found associated with these promoters so that we could then test the proportions of involvement. These 2 gene lists were then analyzed for interaction with the NPC ATACseq and H3K27ac ChIPseq peaks with the utilization of GREAT [43]. GREAT took as input the bed files containing the start and end of the epigenetic peaks and reported the single nearest gene to that peak for each peak in the file. The grep command was then used to find the number of genes in the ASD and non-ASD gene list that were found at least once to be the nearest gene to an epigenetic peak. Fisher’s exact test was used, as described below, to compare the proportion of presence to absence of genes associated with a peak between the ASD genes and non-ASD genes. For other analyses involving this gene list and the variants, the grep command was used to find ASD genes that were annotated as the gene nearest the promoter variant.

#### 4.1.3. Testing for Variants Found in ATACseq and H3K27ac Peaks

ATACseq and H3K27ac peak bed files for human embryonic stem cells and neuronal progenitor (N2) cells were collected from Wu et al., who used the same protocol to generate the cells in the MPRA infection [19], listed under GEO accession GSM517439. For the presence or absence in a peak, the variant bed positions were intersected with the peak bed positions using the bedtools intersect command [42], with the -u parameter being used to obtain only the number of variants intersecting any of the peaks. Likewise, we ran the command a second time with the -v parameter instead of -u to obtain the number of variants that did not intersect with any of the peak positions. We conducted this analysis for at least 2 different conditions and could take the variants found and not found in 2 groups, i.e., high-confidence variants and non-high-confidence variants, and form a 2 × 2 table, with rows indicating the condition and columns indicating the number of variants either found or not found intersecting with a peak. We then took this 2 × 2 file and conducted a Fisher’s test, as described below.

#### 4.1.4. Fisher’s Tests

For variant annotation, presence or absence from a category for 2 conditions of variants were counted and the results were made into a 2 × 2 table that was read into R. In the 2 × 2 table, the double positive number, such as being found involved in a category and a high-confidence variant, was placed in the upper left corner, while the double negative number, such as not being involved in a category and being a non-high confidence variant, was placed in the lower right. This was to allow for correct calculation of the odds ratio. We then ran a base R Fisher’s exact test on the table to compute *p*-value and odds ratio.

### 4.2. MPRA Library Design

#### DNV Selection


Subsetting ~250,000 DNVs from An et al., 2018 [7]


Appendix A from An et al., 2018 [7], contains the list of 6787 promoter DNVs, along with the DNVs’ involvement in several selected annotation categories. Promoter variants are described as those falling within 2000 base pairs of a transcription start site. For the selection of DNVs that are in conserved loci, we subsetted the variants to only those that had a 1 in the annotation conservation_all.


Positive and negative sequence selection


Four ENCODE control regions previously validated by luciferase assays were selected: two negative, two positive (hg19 coordinates).
Negative 1 (chr2:238,336,485–238,336,655)Negative 2 (chr7:96,637,215–96,637,385)Positive 1 (chrX:55,041,354–55,041,524)Positive 2 (chr6:10,147,166–10,147,336)

A total of 148 positive sequences were selected by their overlap with the top 1% quantile of H3K27ac peaks of promoters with TPM ≥ 2 from RNA-seq data generated in N2 cells [36] and ranked by their ATAC-seq peak scores. In detail, these 148 positive controls were selected using the following procedure:

Using H3K27ac, ATAC-seq, and RNA-seq data from N2 cells,

(1)we chose the top 5000 H3K27ac peaks.(2)From them, we chose the max. ATAC-seq peak within each H3K27ac peak.(3)We chose the position of each peak located between 150 and 2000 bp from the TSS transcript.(4)We chose the transcripts with high RNA-seq expression in N2 (RPKM ≥ 2).(5)We obtained a 200-bp sequence based on the midpoint of each peak.

The 150 negative control sequences are scrambled sequences that were generated by randomly choosing 150 test sequences, 75 from cases and 75 from controls, and shuffling all their 200 nucleotides.

### 4.3. Library Processing: Replicates, Association, Barcode Count

#### 4.3.1. Association

Reads from the association library were aligned to the reference set of sequences using bowtie2 [40] with the “very sensitive” preset parameters for maximal accuracy. A barcode was confidently assigned to a sequence if at least 3 unique UMIs supported that assignment and at least 80% of the UMIs associated with that barcode were aligned to the sequence. Barcodes that were not confidently assigned were considered ambiguous and discarded from downstream analyses. Overall, 1,869,972 barcodes were observed, of which 879,815 (47%) were confidently assigned, averaging 125 barcodes per sequence (Appendix A).

#### 4.3.2. MPRA Barcode Counting

Reads from the MPRA libraries were processed against the set of confidently assigned barcodes, requiring a perfect match. Of the barcodes observed in the MPRA libraries, an average of 61.6% were confidently assigned, 37.4% were ambiguous (observed in the association library but were not confidently assigned), and 0.9% were unobserved in the association library (Appendix A). Only barcodes that appeared in at least two corresponding libraries (DNA and RNA libraries from the same time point and replicate) were included in downstream analyses, resulting in an average of 134.4 barcodes per sequence.

#### 4.3.3. Quantification of Induced Transcription Rate with MPRAnalyze

The quantification of induced transcriptional rates (“alpha” values) was performed using MPRAnalyze [29]. Briefly, MPRAnalyze fits two nested generalized linear models (GLMs): the first estimates the latent construct counts from the observed DNA counts, and the second estimates the latent rate of transcription from the latent construct estimates and observed RNA counts. The models are optimized using likelihood maximization, with a gamma likelihood for the DNA counts and a negative binomial likelihood for the RNA counts. MPRAnalyze includes library-size normalization factors, which were computed once using the entire dataset and then used across all analyses, to maintain consistency. For the quantification of alpha values, the full experimental design was included in the design matrix for the DNA model, and an alpha value was extracted for each replicate RNA model.

#### 4.3.4. Classification of Active Sequences with MPRAnalyze

The classification of active sequences was performed using the standard MPRAnalyze classification analysis, in which alpha values are mad-normalized (a median-based variant of z-normalization) and each value is tested against the null distribution, estimated from the alpha values from the negative control scrambled sequences.

### 4.4. MPRA Differential Expression Analysis

Fasta files were read into the MPRAflow [20] program using the parameters: --cigar 200 M --min-frac 0.7 --mapq 1. This tells the program that the fasta file reads should be full length (cigar 200M) and have a minimum fraction of the barcode read mapping to the candidate sequence, plus a barcode association sequence (min-frac 0.7), and that the mapping quality of the read should be set to one, since the candidate sequences are in pairs with only 1 base pair difference, which lowers the mapping quality (mapq 1). The output of the association command is then fed into the count command of MPRAflow using the parameters --bc-length 15 --merge-intersect TRUE --mpranalyze. This will only count barcodes that are of full length (bc-length 15), keep only barcodes found in both the DNA reads and RNA reads (merge-intersect TRUE), and tell the program to format the output to be the input for MPRAnalyze (--mpranalyze).

### 4.5. Testing Motif Interaction with Fimo

We used the program fimo [33] to find transcription factor DNA binding motifs that are involved at the sites that our variants occur in. Adapted from scripts used in Kreimer et al., 2019 [44], we supplied fimo with a 200-bp genomic region surrounding our 1 bp variant site and ENCODE’s motif.meme file to generate a file that listed motifs involved with our variants and the *p*-value for the interaction. We ran this command once with the reference allele in the sequences and once with the alternate allele so that we could compare the motifs that were gained or lost in the allele change. We then filtered the results to include only those that had a *p*-value of less than 1.0 × 10^−5^, and the motif included the 100th bp of the genomic read, meaning that the motif had the allele in question in its sequence. We compared the number of motifs when looking at reference and alternate alleles. Additionally, for each variant, we examined which motifs were found in each run and found which motifs were unchanged (found with both alleles), disrupted (found with reference allele but not with alternate allele), and gained (not found in reference allele but found with alternate allele).

### 4.6. Finding Cell Process and Protein–Protein Interaction Enrichment Using GSEA and DOMINO

The gene set enrichment analysis (GSEA) webpage interface was used to find cell process enrichments in our datasets, specifically the Investigate Gene Sets tab of the Molecular Signatures Database. We pasted the Official Gene Name into the input box with one gene name per line and clicked to use the GTEx compendium as the expression profiles. We then computed the overlaps with the following gene sets chosen: CP (canonical pathways), TFT (all transcription factor targets), and C5 (ontology gene sets). We then changed the setting to show the top 100 enriched gene sets and computed with the default FDR q-value of less than 0.05.

We utilized the web interface of DOMINO and uploaded a text file containing a list of the Official Gene Names with one name per line. We then selected the option to use the PCNet [45] network file to compute protein–protein interactions. For each subnetwork that DOMINO provided, we ordered the enriched processes by *p*-value and analyzed all of the significantly enriched processes for all subnetworks of our gene set.

### 4.7. Experimental Procedures

#### 4.7.1. Preparation of N2 Cells

Neural induction of H1 hESCs (WiCell WA-01, RRID:CVCL_9771) was initiated with noggin, a BMP inhibitor, and cells were cultured in retinoic acid-free media supplemented with growth factors FGF and EGF in order to generate early neural progenitors (N2; passage 15) following the protocol used by Wu et al. [19].

#### 4.7.2. ChIP-seq of H3K27ac in N2 Cells

ChIP-seq was performed using a LowCell# ChIP kit (Diagenode, Denville, NJ, USA) according to the manufacturer’s instructions with modifications. Briefly, cultured N2 cells in 10 cm dishes were crosslinked in 1% formaldehyde (Thermo Fisher Scientific, Waltham, MA, USA) for 5 min. Crosslinking was quenched with 125 mM glycine. The cells were washed with PBS and precipitated with centrifugation at 6000 rpm for 5 min. The cell pellet was stored at −80 °C for each time point, so that all the samples were processed together. The pellet was lysed in 250 μL of Buffer B (LowCell# ChIP kit) supplemented with complete protease inhibitor (Roche, Basel, Switzerland) and 20 mM Na-butyrate (Sigma-Aldrich, St. Louis, MO, USA). A measure of 130 μL of lysed chromatin was sheared using a Covaris S2 sonicator to obtain on average 250 bp-sized fragments. A measure of 870 μL of Buffer A (LowCell# ChIP kit) supplemented with complete protease inhibitor (Roche) and 20mM Na-butyrate (Sigma) was added to the shared chromatin. A measure of 20 μL of the chromatin solution was saved as an input control. To obtain magnetic bead-antibody complexes, a mixture of 40 μL of Dynabeads protein A and 40 μL of Dynabeads protein G was washed twice with Buffer A (LowCell# ChIP kit) and resuspended in 800 μL of Buffer A. A measure of 10 μg of H3K27ac antibodies (Abcam Cat# ab4729) were added to the washed beads and gently agitated at 4 °C for 2 h. The beads-antibody complex was precipitated with a magnet and the supernatant was removed. A measure of 800 μL of shared chromatin was added to the beads-antibody complex and rotated at 4 °C overnight. The immobilized chromatin was then washed with Buffer A three times and Buffer C once, and eluted in 100 μL of IPure elution buffer (IPure kit; Diagenode). In addition, 80 μL of IPure elution buffer was added to the 20 μL of input that was saved before immunoprecipitation and the DNA was purified using the IPure kit. Purified DNA was sheared using a Covaris S2 sonicator once again to obtain on average 250-bp fragments. Sequencing libraries were generated using the ThruPLEX DNA-seq kit (Rubicon Genomics, Mountain View, CA, USA) according to the manufacturer’s protocol. The DNA was size selected using SPRIselect (Beckman Coulter, Brea, CA, USA). Volumes of 0.7× and 0.9× of SPRIselect were used for right side and left side selection, respectively. DNA was quantified with a Qubit DNA HS assay kit and a Bioanalyzer using the DNA High Sensitivity kit (Agilent Technologies, Santa Clara, CA, USA). Massively parallel sequencing was performed on an Illumina HiSeq4000 with a 50-bp single-end read. ChIP-seq was conducted with two biological replicates.

#### 4.7.3. ATAC-seq in N2 Cells

ATAC-seq was performed according to previously described protocol [46] with modifications. Briefly, 50,000 cells were dissociated using Accutase and precipitated with centrifugation at 500× *g* for 5 min. The cell pellet was washed with PBS, resuspended in 50 μL lysis buffer (10 mM Tris·Cl, pH 7.4, 10 mM NaCl, 3 mM MgCl_2_, 0.1% Igepal CA-630), and precipitated with centrifugation at 500× *g* for 10 min. The nuclei pellet was resuspended in 50 μL transposition reaction mixture, which included 25 μL Tagment DNA buffer (Nextera DNA sample preparation kit; Illumina, San Diego, CA, USA), 2.5 μL Tagment DNA enzyme (Nextera DNA sample preparation kit; Illumina), and 22.5 μL nuclease-free water, and incubated at 37 °C for 30 min. Tagmented DNA was purified with the MinElute reaction cleanup kit (QIAGEN). The DNA was size selected using SPRIselect (Beckman Coulter) according to the manufacturer’s protocol. Volumes of 0.6× and 1.5× of SPRIselect were used for right and left side selection, respectively. Library amplification was performed, as has been previously described [46]. The amplified library was further purified with SPRIselect as described above. DNA was quantified on a Bioanalyzer using the DNA High Sensitivity kit (Agilent). Massively parallel sequencing was performed on an Illumina HiSeq2500 or HiSeq4000 with PE150. ATAC-seq was conducted in 2 biological replicates.

### 4.8. LentiMPRA Library Cloning and Sequence–Barcode Association

The lentiMPRA library construction was performed as previously described [20]. In brief, the array-synthesized oligo pool was amplified by 5-cycle PCR using forward primer (5BC-AG-f01, Appendix A) and reverse primer (5BC-AG-r01, Appendix A) that add mP and spacer sequences downstream of the sequence. The amplified fragments were purified with 1.8× AMPure XP (Beckman coulter) and proceeded to second round 11-cycle PCR using the same forward primer (5BC-AG-f01) and reverse primer (5BC-AG-r02, Table Appendix A) to add a 15-nt random sequence that served as a barcode. The amplified fragments were then inserted into the *Sbf*I/*Age*I site of the pLS-SceI vector (Addgene, Watertown, MA, USA, 137725) using the NEBuilder HiFi DNA Assembly mix (NEB, Ipswich, MA, USA), followed by transformation into 10beta competent cells (NEB, C3020) using the Gemini X2 machine (BTX). We note that there was not a typical polyA signal downstream of the WPRE in our lentiviral vector, as it has been reported that an internal polyA signal can decrease virus titer [47]. Colonies were allowed to grow up overnight on carbenicillin plates and were midiprepped (Qiagen, Redwood City, CA, USA, 12945). We collected approximately 1 million colonies, so that on average, 100 barcodes were associated with each sequence. To determine the sequences of the random barcodes and their association to each sequence, the sequence-mP-barcodes fragment was amplified from the plasmid library using primers that contain flowcell adapters (P7-pLSmP-ass-gfp and P5-pLSmP-ass-i#, Appendix A). The fragment was then sequenced with a NextSeq 150PE kit using custom primers (R1, pLSmP-ass-seq-R1; R2 (index read), pLSmP-ass-seq-ind1; R3, pLSmP-ass-seq-R2, Appendix A) to obtain approximately 50 M total reads.

### 4.9. Lentiviral Infection and Barcode Sequencing

Lentivirus was produced in twelve 15-cm dishes of 293T cells using a Lenti-Pac HIV expression packaging kit following the manufacturer’s protocol (GeneCopoeia, Rockville MD, USA, LT002). Lentivirus was filtered through a 0.45 um PES filter system (Thermo Scientific, 165-0045) and concentrated using a Lenti-X concentrator (Takara Bio, San Jose, CA, USA, 631232). Titration of the lentiMPRA library was conducted on human NPCs (N2s), as described previously [20]. Lentiviral infection, DNA/RNA extraction, and barcode sequencing were all performed as previously described [21]. Briefly, approximately 1 million N2 cells (three 10-cm dishes) were infected with the lentivirus library with a multiplicity of infection (MOI) of 50, along with 8 µg/mL polybrene (Sigma). Three independent replicate cultures were infected. The cells were incubated for 4 days with a daily change of the media. DNA and RNA were purified using an AllPrep DNA/RNA mini kit (Qiagen). RNA was treated with Turbo DNase (Thermo Fisher Scientific) to remove contaminating DNA, and reverse transcribed with SuperScript II (Invitrogen, 18064022) using barcode-specific primer (P7-pLSmp-assUMI-gfp, Appendix A), which has a unique molecular identifier (UMI-10 bp index). Barcode DNA/cDNA from each replicate were amplified with 3-cycle PCR using specific primers (P7-pLSmp-assUMI-gfp and P5-pLSmP-5bc-i#, Appendix A) to add sample index and UMI. A second round of PCR was performed for 19 cycles using P5 and P7 primers (P5, P7, Appendix A). The fragments were purified and further sequenced with NextSeq 15PE with 10-cycle dual index reads, using custom primers (R1, pLSmP-ass-seq-ind1; R2 (index read1 for UMI), pLSmP-UMI-seq; R3, pLSmP-bc-seq; R4 (index read2 for sample index), pLSmP-5bc-seqR2, Appendix A).

## Figures and Tables

**Figure 1 ijms-24-03509-f001:**
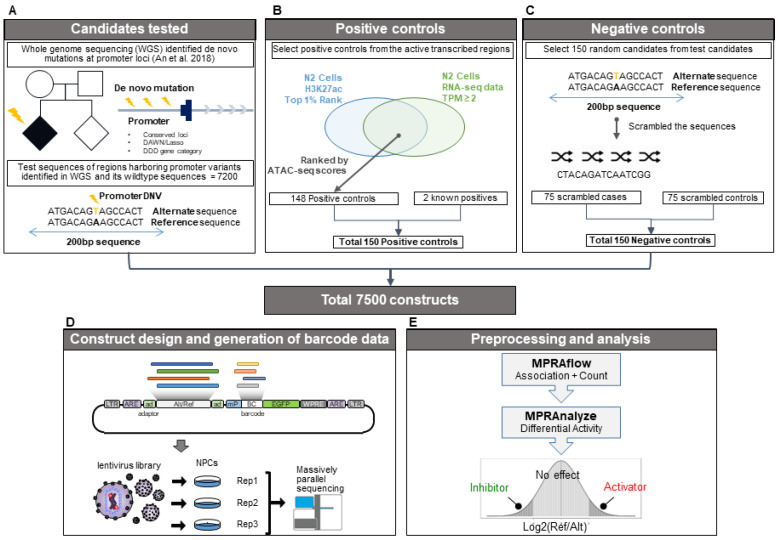
**LentiMPRA analysis pipeline.** (**A**) Selection of the 7200 candidate regulatory sequences (CRS) generated from the reference and alternate allele sequences of 3600 variants. (**B**) Selection process of the top active regions in N2 cells used to generate the positive control sequences. (**C**) Selection process for generating the negative control sequences. (**D**) Diagram of the lentiviral vector used for infection into our three replicate NPC cell populations. (**E**) Diagram of the computational pipeline used to associate and count the randomly added CRS and barcode sequences to be used for differential expression analysis [7].

**Figure 2 ijms-24-03509-f002:**
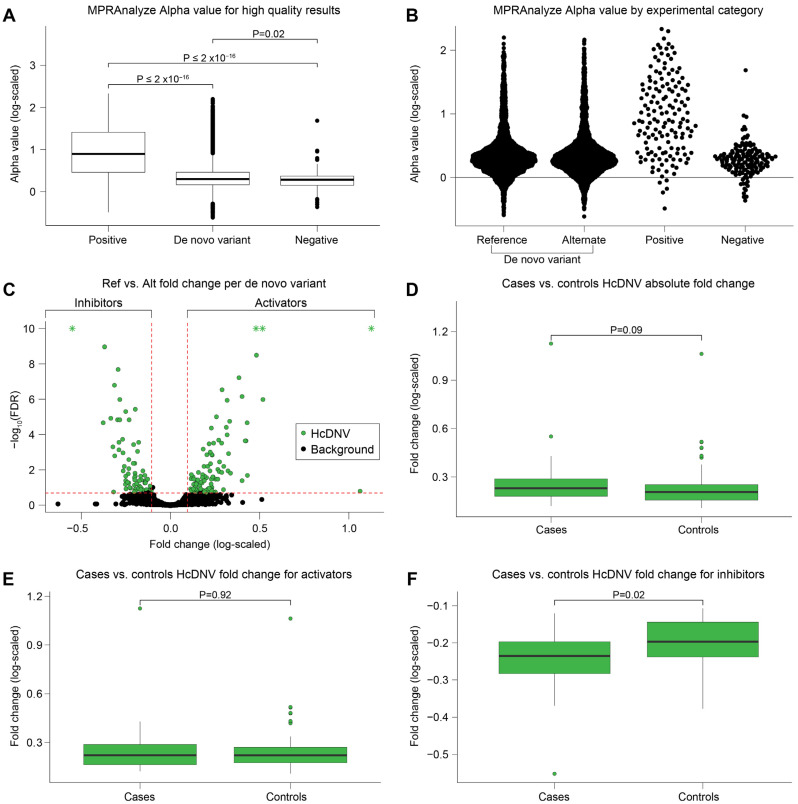
**MPRAnalyze results.** (**A**) Boxplot comparing the log of the alpha value reported by MPRAnalyze. The *p*-values above the boxes are Wilcoxon rank sum test *p*-values comparing the alpha value between the boxes indicated by the end of the brackets. (**B**) Swarm plot of the log alpha values reported in (**A**) showing the density and distribution of the values. (**C**) Volcano plot showing the natural log fold change between the alternate allele alpha value over the reference allele alpha value on the *x*-axis and the –log10 of the MPRAnalyze FDR reported for the differential expression on the *y*-axis. Y-axis values greater than 10 have been lowered to 10 and marked with an “*” to zoom in the graph. (**D**–**F**) Boxplots comparing the log fold change values between HcDNV cases and controls for the absolute value of the log fold change (**D**), activators only (**E**), and inhibitors only (**F**).

**Figure 3 ijms-24-03509-f003:**
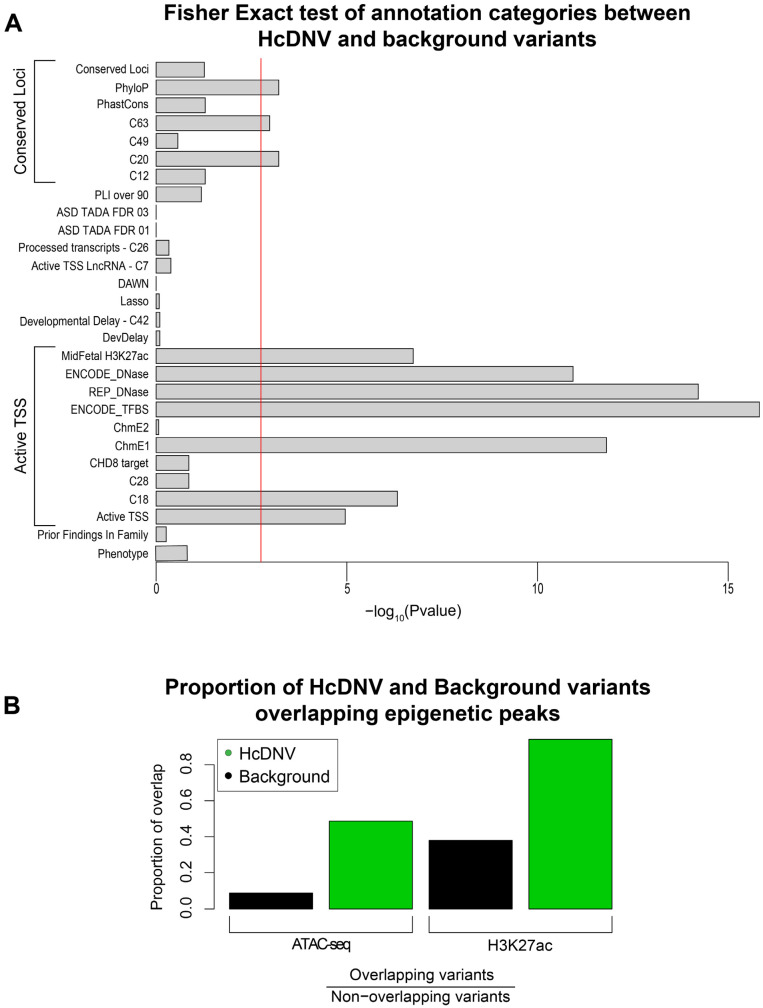
Characterization of variant involvement in regulatory and epigenetic regions. (**A**) Barplot of Fisher’s exact *p*-values comparing the proportion of HcDNV and background variants found in each annotation category. The red line indicates the Bonferroni multiple testing-corrected *p*-value threshold. (**B**) Barplot of the proportion of overlap of HcDNV and background variants with H3K27ac and ATACseq peaks. ATACseq *p*-value = 5.4 × 10^−18^; H3K27ac *p*-value = 4.0 × 10^−8^.

**Figure 4 ijms-24-03509-f004:**
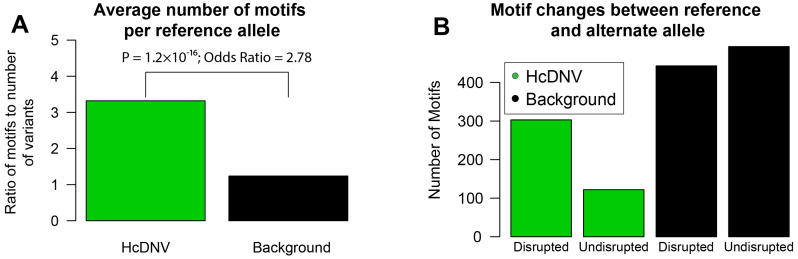
**Transcription factor binding motif disruption in HcDNV variants.** Fimo was used to call transcription factor binding motifs that overlapped with the variant position. Fisher’s exact test was used to find the *p*-value and odds ratio for the proportions in (**A**,**B**). (**A**) Barplot showing the ratio of the number of filtered motifs found over the number of variant sequences used to find overlapping motifs. (**B**) Barplot showing the number of filtered motifs for HcDNV and background variants separated into those that are found only with the reference allele version of the CRS and not the alternate allele version (Disrupted), and motifs that are found in both the reference and alternate forms of the CRS (Undisrupted). Fisher’s exact test *p*-value: 4.9 × 10^−30^, Odds Ratio: 2.68.

**Figure 5 ijms-24-03509-f005:**
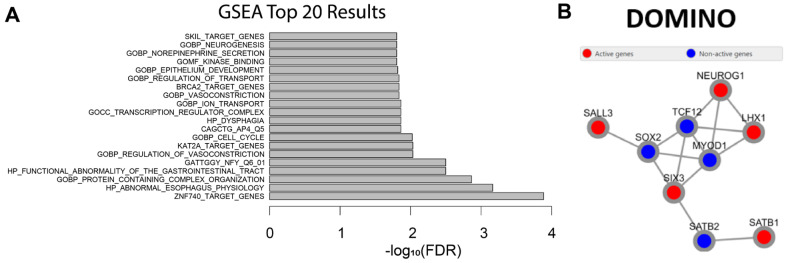
**Gene and protein interaction of genes affected by variants**. (**A**) GSEA was used to find enriched processes among the genes affected by the HcDNVs. Y-axis indicates the top 20 enriched categories and the x-axis indicates the −log10 of the FDR scores. (**B**) DOMINO was used to find subnetworks of protein–protein interactions and the cellular processes that are enriched for those genes. One significant network is shown above where the red active nodes are genes affected by the HcDNVs and the blue inactive nodes are intermediate proteins added by DOMINO. The top five enriched processes are: DNA-binding transcription factor activity RNA polymerase II specific, RNA polymerase II transcription regulatory region sequence specific DNA binding, DNA-binding transcription factor activity, and transcription regulatory region nucleic acid binding.

**Figure 6 ijms-24-03509-f006:**
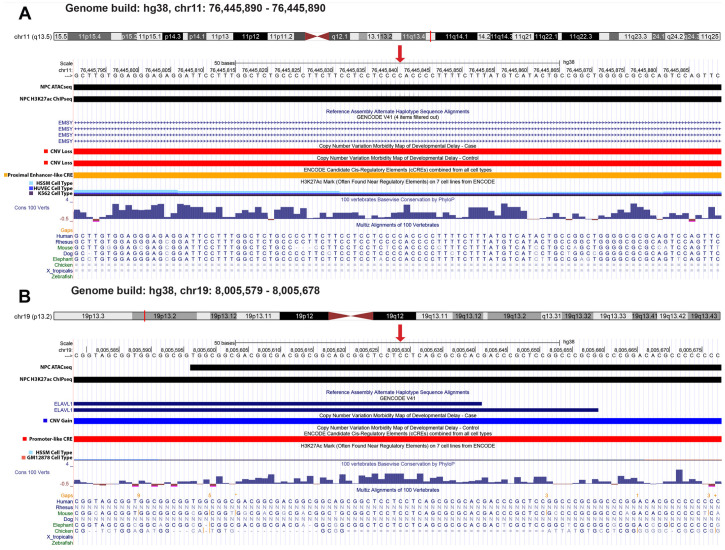
GC content of HcDNV variants and top scoring variants. (**A**,**B**) Genome browser capturing the variant locations (denoted by the red arrow) of variants affecting EMSY (**A**) and ELAVL1 (**B**). Track names are on the left with custom tracks NPC_ATAC and NPC_H3K27ac to show the ATACseq and H3K27ac ChIP-seq peak data that were gathered for this cell type.

## Data Availability

Raw and processed files can be found in GEO under accession number: GSE216129.

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
