# Peer review of "Characterization of De Novo Promoter Variants in Autism Spectrum Disorder with Massively Parallel Reporter Assays"

_ijms, 2023, doi:10.3390/ijms24043509_

Round 1

Reviewer 1 Report

The aim of the paper was to characterise de novo promoter variants in patients with autism spectrum disorder (ASD) and their healthy siblings. Massively parallel reported assays were utilised to detect transcriptional consequences of the de novo variants in neural progenitor cells. This was an interesting approach and shows promise as a reliable model.

The article is well written and the research design was appropriate. The methods were adequately described and the results clearly presented. The conclusions supported the results and the limitations of the approach were clearly stated.

This research involved a significant amount of work, unfortunately the results did not identify significant enrichment for de novo variants in ASD cases or evidence of increased functional impact. The authors should consider addressing some of the limitations of this research as outlined in the discussion section.

Reviewer 2 Report

Koesterich and colleagues present an interesting manuscript in which the authors attempt to link the functions of DNVs to ASD. The manuscript has merit and can provide interesting conclusions about the role of DNVs in the etiology of ASD. However, I have several concerns that limit my enthusiasm for the presented work. My comments are highlighted below in the order they first occurred perusing the manuscript. 

1. My first comment is that the authors did not, in fact, find evidence that DNVs are implicated in ASD. To that end, all subsequent analyses seem redundant.

2. My second comment is that the authors did not identify any significant result! Their choice of selected FDR<0.2 is relatively lax and did not explain why they used this threshold for multiple testing. It can easily be argued that these findings are stochastic in nature. 

2. My third comment is that although rare variants generally have larger effects, the methodology used to introduce these (i.e., one by one) may not be the best way to test for their impact on gene expression. While I understand their rationale for the approach they used, another possibility would have been to introduce these variants based on their predicted functions as an aggregate and test how they will affect transcription.

Round 2

Reviewer 2 Report

The authors have attempted to address the reviewers' concerns; however, in my opinion, these are still inadequate. Mainly they state that "...we were unable to find significant evidence to implicate these DNVs with ASD..., and we are transitioning from ASD implication to significant variants reported by MPRA involvement in transcription we state at the end of 2.4 on lines 180-182 and the beginning of 2.5 on lines 195-197 that the following analyses are separate from the negative results of the ASD implication and therefore we can still find significant findings." In my opinion, though, this attempt is insufficient. Unfortunately, adding a few sentences here and there is not enough. The manuscript still appears to have identified significant ASD findings, which is misleading. Ideally, the authors need to restructure the manuscript by focusing on using MPRA as a methodological approach to identify rare variants, i.e., they need to present this manuscript as a rather methodological paper.        

Round 3

Reviewer 2 Report

I have no further comments to the authors